# Bidirectional Response of Weak-Axis End-plate Moment Connections: Numerical Approach

**Eduardo Nuñez** [1,]*[ ], **Guillermo Parraguez** [2] **and Ricardo Herrera** [2][ ]

1   Department of Civil Engineering, Universidad Católica de la Santísima Concepción, Concepción 78349, Chile
2   Department of Civil Engineering, Universidad de Chile, Santiago de Chile 8370449, Chile; guillermo.parraguez@ug.uchile.cl (G.P.); riherrer@ing.uchile.cl (R.H.)
*   Correspondence: enunez@ucsc.cl; Tel.: +56-9-5127-7382

**Abstract:** Brittle failure mechanisms can affect the seismic performance of structures composed of intersecting moment resisting frames, if the biaxial effects are not considered. In this research, the bidirectional cyclic response of H-columns with weak-axis moment connections was studied using numerical models. Several configurations of joints with bidirectional effects and variable axial loads were studied using the finite element method (FEM) in ANSYS v17.2 software. The results obtained showed a ductile behavior when cyclic loads are applied. No evidence of brittle failure mechanisms in the studied joint configurations was observed, in line with the design philosophy established in current seismic provisions. However, beams connected to the column minor axis reached a partially restrained behavior. Joints with four beams connected to the column exhibited a partially restrained behavior for all axial load levels. An equivalent force displacement method was used to compare the hysteretic response of 2D and 3D joints, obtaining higher deformations in 3D joints with respect to 2D joints with a similar number of connected beams. Consequently, design procedures are not capable of capturing the 3D deformation phenomenon.

**Keywords:** moment connection; finite element method; weak-axis bending; hysteretic behavior

## 1. Introduction

The steel moment frame is a structural system commonly used as an alternative in steel buildings. Horizontal members (beams) and vertical members (columns) are joined by means of fully or partially restrained moment connections. The lateral resistance is provided by flexural and shear mechanisms in beams and columns, capable of reaching high ductility levels. In these systems, moment connections play an important role in their seismic performance and they may be designed according to [1], which includes seismic design requirements for connections commonly used in beam-to-column joints with wide flange columns and beams. In several parts of the world, the design practice calls for the use of every frame elevation to resist the seismic loads, which generates columns that are part of two seismic resisting systems oriented in different directions. However, the cyclic response of moment resisting joints in columns subjected to bidirectional effects has not been extensively studied. The latest version of the seismic provisions [2] provides general guidelines to design the columns, when they are part of two intersecting special moment frames in different directions, such as considering the possible effects of yielding of the beam framing into the column in both directions simultaneously. However, no prequalified connection framing on the web of wide flange columns is available in [1]. Furthermore, clearly defined strong-column/weak-beam criteria are only available for planar frames to provide for wide flange columns strong enough and capable of distributing yielding over multiple stories [2]. On the other hand, brittle failure mechanisms can affect the seismic performance of the structures if the

biaxial effects are not considered, according to [3]. An overview of the previous research on weak axis connections is presented below.

The cyclic response of a reduced beam section moment connection to the weak axis of the column was studied by [4]. In this research, two full-scale tests were performed. The results obtained showed a ductile behavior of weak-axis Reduce beam section (RBS) connections, reaching 0.03 (rad) plastic rotation. However, field welding is required in the RBS moment connection. A few years later, research conducted by [5] studied a new weak-axis moment connection improving constructability details. Three specimens were tested under a monotonic load, reaching high deformation levels.

Experimental research on wide flange beams, connected to wide flange columns, with large width–thickness ratios, subjected to a cyclic load about the weak-axis was performed [6]. A failure mechanism controlled by local buckling in all specimens was obtained, demonstrating the strong influence of the flange/web and width–thickness relationship. Shim. et al. [7] conducted an experimental study of a new weak-axis moment connection. Bolted splices at the top and bottom flange and web were considered. The results showed an improvement in the structural performance of weak-axis moment connections, allowing the removal of typical brackets in this type of connection.

Oh. et al. [8] evaluated the seismic behavior of column-tree moment connections with reduced flanges connected to the weak-axis of columns in steel moment frames. A ductile behavior was obtained in the tested specimens, which reached a 0.05 (rad) story drift ratio. The requirements established in the seismic provisions were satisfied. An evaluation of seismic performance in weak-axis moment connections with high-tension bolts was developed by [9]. The significant effect of high-tension bolts in the cyclic response of weak-axis moment connections was observed. The requirements of partially restrained moment connections defined in design codes and were verified when more than four bolts were employed.

Ying. et al. [10] conducted an experimental study of weak-axis moment connections with I-beams and H-columns. A cyclic load was applied in joint specimens and the influence of composite action in the hysteretic behavior was obtained. The results showed a welding fracture at the conjunction of the diaphragm and beam-end flange. Additionally, a story drift of 0.04 (rad) in composite specimens was reached. Experimental research of cyclic behavior in semi-rigid joints was performed by Shi et al. [11]. In this proposal, T-stubs connecting beams to the weak-axis of the column were used. The results showed failure in beams and T-stub fractures. Furthermore, a decrease in rotational stiffness with an increase in the end-plate thickness was obtained.

In this research, a numerical study of the bidirectional cyclic response of weak-axis beam-to-column joints using end-plate moment connections was conducted. Several configurations of joints with bidirectional effects and variable axial loads were studied using the finite element method (FEM) in ANSYS software [12].

## 2. Weak-Axis Moment Connection Design

In this research, an extended unstiffened end-plate with four bolts was studied to connect I-beams with H-columns, considering the bidirectional and axial load effects. In this type of moment connection, beams connected to the weak and strong axes of a column were considered simultaneously. The end-plates were bolted to the column using high-strength bolts and horizontal and vertical diaphragms were welded to the H-column on the weak axis connection (see Figure 1).

Additionally, complete joint penetration (CJP) welds were employed between the diaphragms and the column. In addition, fillet welds were used between the beam web and the end plate. The selected configuration improves the field erection process, eliminating field welding. Furthermore, four configurations of beam-column joints were analyzed, considering the bidirectional cyclic response and different axial load levels.

The size of beam, end-plate and column were obtained according to [2]. First, the seismic design of a low-rise building located in Santiago, Chile, with steel moment frames and a story height of 3.50 (m) was performed, according to [2]. The elements of the connection were designed following

the procedure established in [1]. The vertical diaphragms were designed for the expected shear force, and the horizontal diaphragms were designed for the expected tension of the beam flange. Figure 2 shows the elements, dimensions and details of the connection. The moment capacity/beam capacity ratio in the joints was calculated uniquely in the strong-axis of the column, considering the contribution of beams connected in the weak-axis, according to [2].

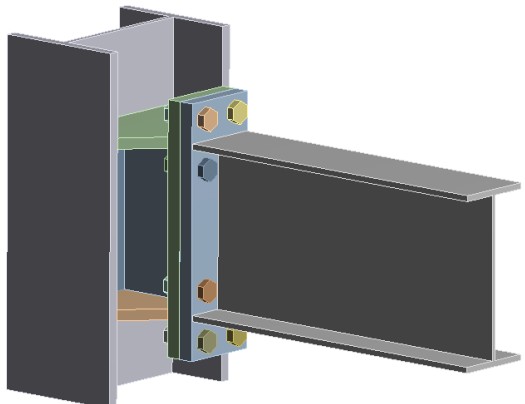

**Figure 1.** View of a weak-axis moment connection.

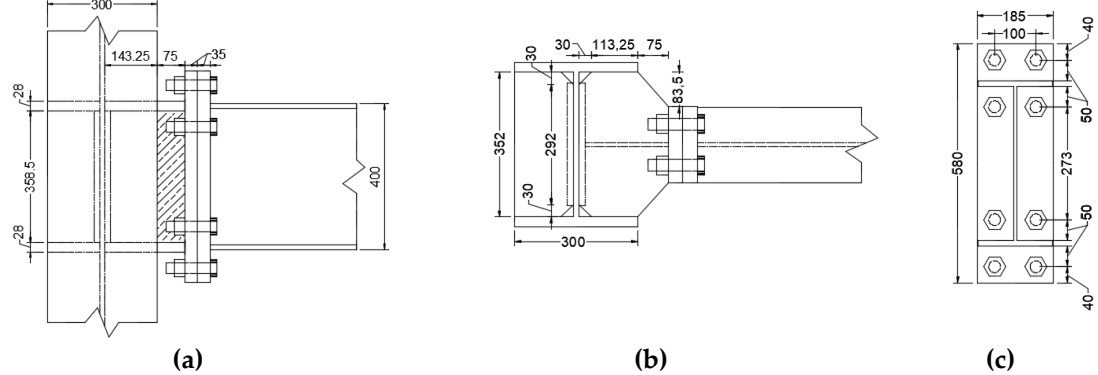

| (a) | (b) | (c) |
| --- | --- | --- |

**Figure 2.** (**a**) Elevation view of moment connection, (**b**) plan view of moment connection, and (**c**) details of the distance between end-plate bolts (all dimensions in mm).

## 3. Numerical Study

In this research, the cyclic behavior of weak-axis moment connections was studied the using finite element method with ANSYS Software [12]. Different 3D and 2D joint configurations of beam column joints, considering variability of the axial load and bidirectional effects simultaneously, were analyzed. In Figure 3, a 3D view of the joint configurations is shown and the simulation matrix showing the different axial load levels is reported in Table 1. Appropriate materials, geometrics, contact nonlinearities, and boundary conditions were used. Additionally, interior and exterior joints were studied, assuming representative inflection points in columns and beams. In addition, the diameter of bolts and holes were assumed to be similar, the welds were not included in the numerical model and the dimensions of nuts and heads were deemed similar. These considerations demonstrated good results when the end-plate moment connections modeled were subjected to cyclic loads, such as was studied by [13,14].

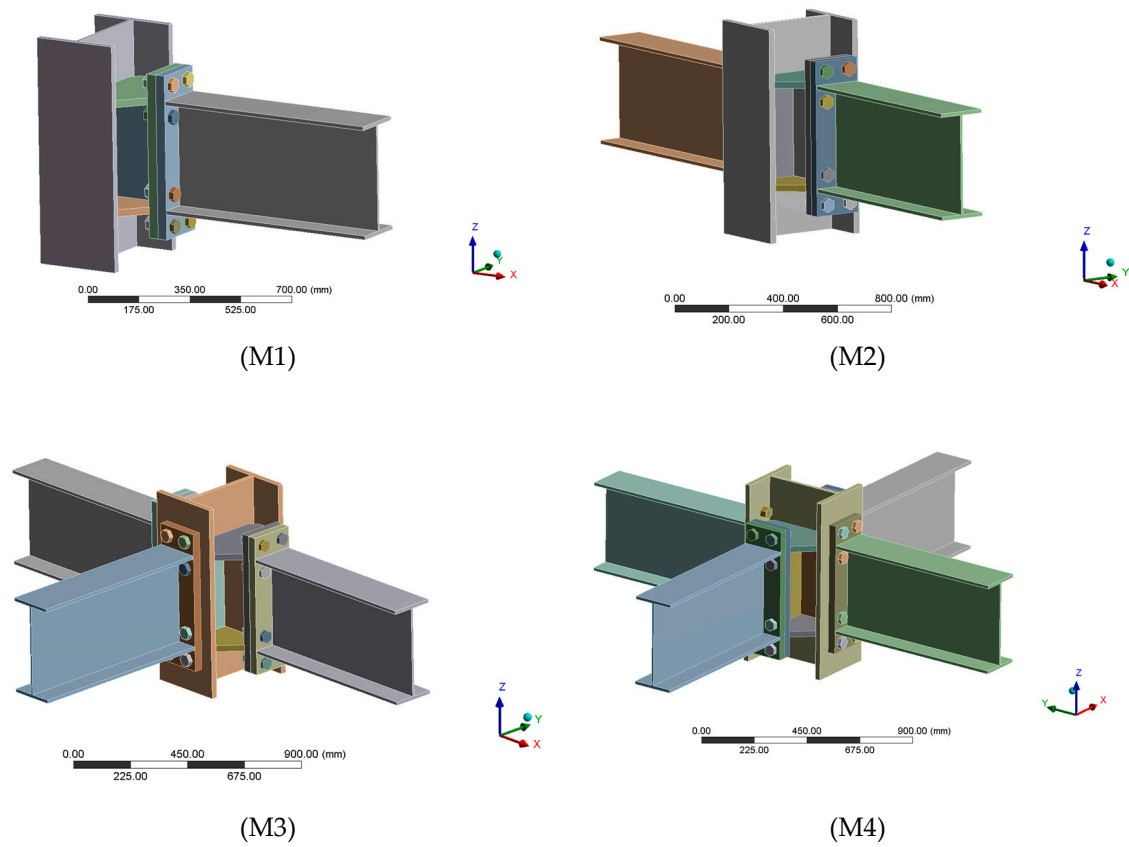

**Figure 3.** Interior (**M2,M4**) and exterior (**M1,M3**) joint configurations studied.

**Table 1.** Configuration of beam-to-column joints in FE models.

| No. | Model | Joint | P/Py |
|---|---|---|---|
| 1 | | M1-00 | 0 |
| 2 | 1 beam (M1) | M1-25 | 25% |
| 3 | | M1-50 | 50% |
| 4 | | M2-00 | 0 |
| 5 | 2 beams–interior (M2) | M2-25 | 25% |
| 6 | | M2-50 | 50% |
| 7 | | M3-00 | 0 |
| 8 | 3 beams–exterior (M3) | M3-25 | 25% |
| 9 | | M3-50 | 50% |
| 10 | | M4-00 | 0 |
| 11 | 4 beams–interior (M4) | M4-25 | 25% |
| 12 | | M4-50 | 50% |

Note: $Py = FyAg$, where $Fy$ is yield stress and $Ag$ is the gross area of the section.

Two types of elements were considered in the numerical model: BEAM188 and SOLID186. The BEAM188 elements with two nodes and 6 degrees of freedom (DOF) simplified the numerical model in zones with elastic behavior. The SOLID186 elements, with 20 nodes and three DOF per node, were used to model beam, column and connection elements because they allow the use of the inelastic behavior of materials, such as plasticity and hardening, large deflections, and contact nonlinearities. The use of SOLID186, an element based on quadratic interpolation functions, was also justified by its adequacy to model regions with non-straight contours, such as the bolt holes or the horizontal diaphragms. In Figure 4, a schematic view of meshing, the types of contacts and the loads applied are shown. Each model was composed of solid elements plus beam elements. These elements were joined

using a multi-point constraint (MPC), as shown in Figure 4d, which establishes compatibility between the six DOFs of the beam element and the displacement DOFs of the solid elements. In SOLID186 elements, a fine mesh was used to capture the large inelastic deformations. The number of nodes per model studied is summarized as follows: M1 (43608), M2 (72877), M3 (113676) and M4 (136366).

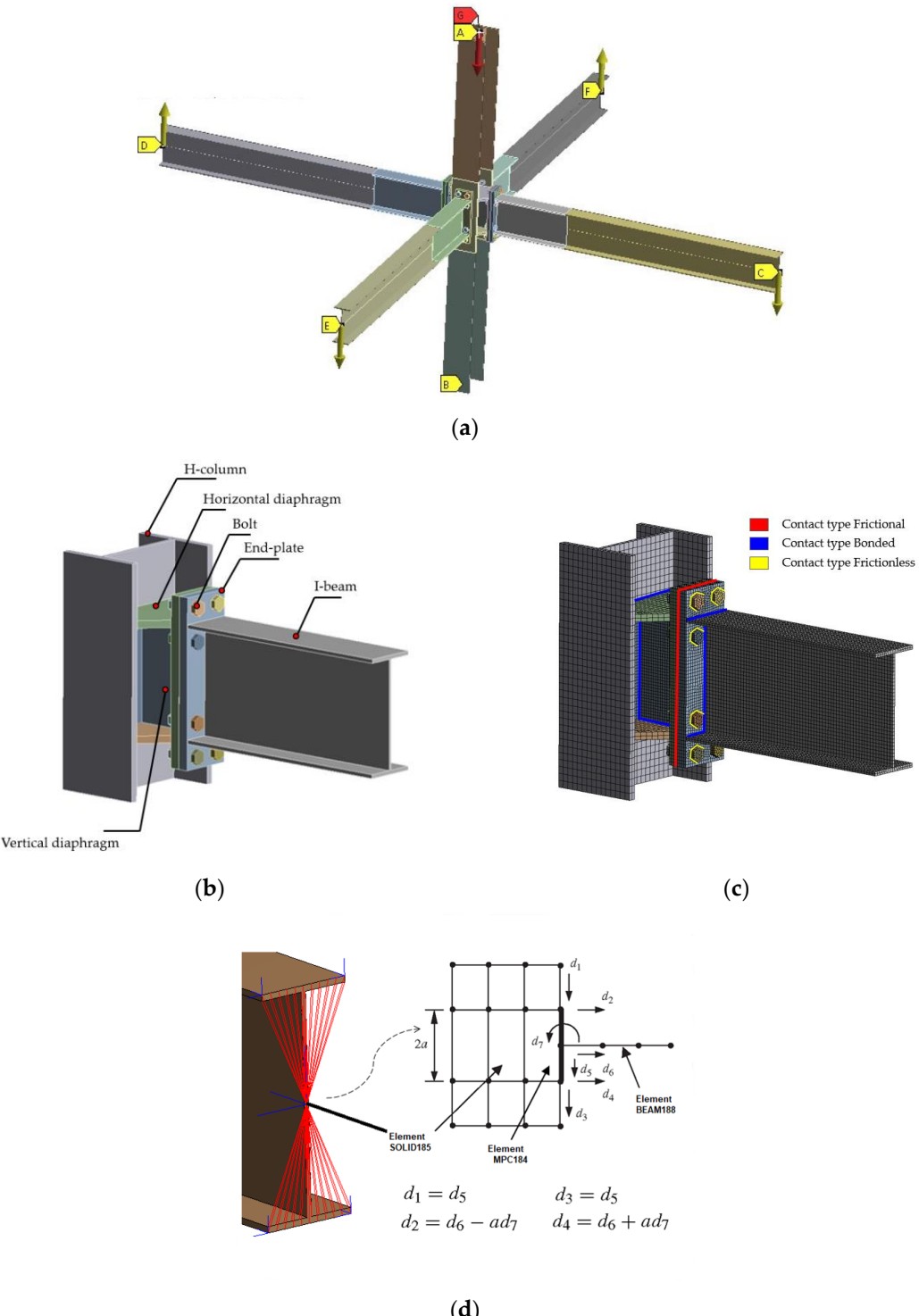

**Figure 4.** (**a**) Boundary conditions, (**b**) elements in moment connection, (**c**) contacts types in numerical model, (**d**) interface between frame element and solid elements.

In Figure 4a, boundary conditions were assigned to FEM models to emulate the conditions that would be applied in actual tests. Two types of supports were considered: pinned supports at the column ends (all three displacements restrained at the base of the column and horizontal displacements at the top), and rollers (out-of-plane displacements restrained) at the end of beams. Furthermore, vertical displacements at the beam ends were applied using the "Remote Point Displacement" command (the Remote Point is a command for remote boundary conditions. Remote points are a way of abstracting a connection to a solid model, be it a vertex, edge, face, body, or node, to a point in space. The solver uses multipoint constraint equations to make these connections, according to [12]. In this study, the displacements were applied by means of a Remote Point command) according to [2]. A bolt pretension was applied as specified in [2]. A "Bonded" type contact was used to simulate all the welds between the diaphragms and the beams and columns, and the beams and end-plates. Previous studies [13] on a connection using similar details, but with Hollow structural sections (HSS) columns, showed no inelastic effects occur in any weld. The type of contact between the bolts, bolt holes, nuts and end-plates was characterized through two types of contacts: a "Frictional" type contact with a friction coefficient equal to 0.3 according to [13], and a "Frictionless" type contact which provided support in the normal direction to the selected element. Furthermore, movements and rotations were restrained in the normal direction, but movements or rotations in a tangential direction with zero friction were permitted, according to [15,16]. The connections were designed to avoid inelastic deformation in the bolts and nuts. Therefore, it was considered unnecessary to model the nut stripping. Regarding the shank necking, the bolt thread is not included in the model, therefore only inelastic deformations in the gross area of the bolt can be captured. However, if these effects need to be considered, previous studies by [17] provide a methodology to include them in the numerical model. Finally, the Incremental Newton–Raphson method was used. In this method, the nonlinearities are considered through the sub-steps for each load step. The Force Convergence Value criterion was used, where the residual out-of-balance force vector [R] = [Fa] − [Finr] and the force convergence value must be below the value for convergence, according to [12]. The Augmented Lagrange method was employed to reach numerical convergence in the contact zone, according to [14]. The contact types and boundary conditions are shown in Figure 4b,c and Table 2 displays the displacements applied according to the protocol established in [2].

**Table 2.** Displacements applied, according to [2].

| No. | Number of Cycles | Drift Angle θ (rad) |
| --- | --- | --- |
| 1 | 6 | 0.00375 |
| 2 | 6 | 0.00500 |
| 3 | 6 | 0.00750 |
| 4 | 4 | 0.01000 |
| 5 | 2 | 0.01500 |
| 6 | 2 | 0.02000 |
| 7 | 2 | 0.03000 |
| 8 | 2 | 0.04000 |

Note: continue, applying increments of θ = 0.01 (rad), with two cycles of loading.

A typical ASTM-A36 material was used in the beams, columns and diaphragms, according to Chilean practice. ASTM-A490 was used to simulate the bolt material. Constitutive law parameters for steel elements were obtained from coupon specimens tested by [18], and their values are reported in Table 3. The inelastic model was formulated using the von Mises yielding criteria and an associated flow rule. The resulting bilinear stress–strain curves are shown in Figure 5.

**Table 3.** Material constitutive law parameters of steel elements, according to [18].

| Element | Designation | $\sigma_y$ (MPa) | $\varepsilon_y$ | $\sigma_u$ (MPa) | $\varepsilon_u$ |
|---|---|---|---|---|---|
| Bolts | ASTM-A490 | 1156 | 0.00586 | 1433 | 0.14 |
| Beam, Column, Horizontal diaphragms, Vertical diaphragms, End-plate | ASTM-A36 | 293 | 0.001465 | 445 | 0.24 |

Note: $\sigma_y$: yield stress; $\varepsilon_y$: yield strain; $\sigma_u$: ultimate stress; $\varepsilon_u$: ultimate strain.

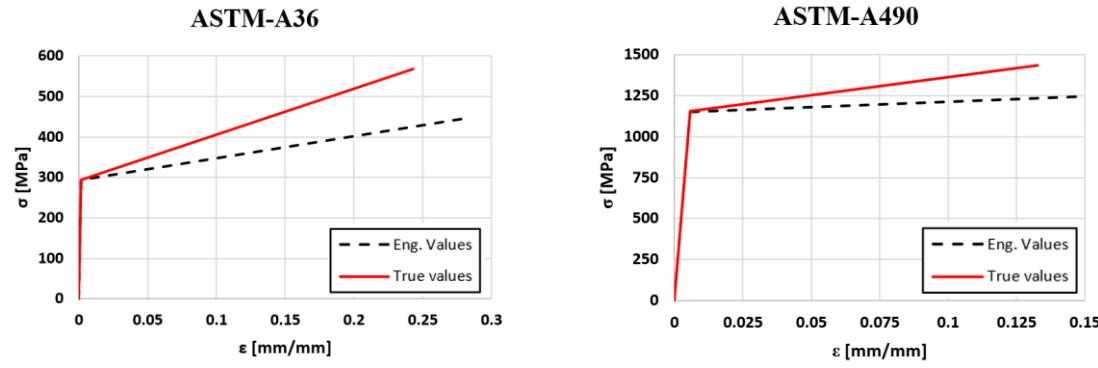

**Figure 5.** Simplified stress–strain relation of materials employed in FE models.

## 4. Analysis of Results

An interstory drift angle of 4% should be accommodated and the flexural resistance of the beam must be at least 80% of the plastic moment capacity of the beam, $M_p$, for moment connections in special moment frames according to [2]. Additionally, a premature failure of columns should be avoided by concentrating the inelastic response in the beams. In the following sections, the results of the cyclic response with bidirectional effects are shown.

### 4.1. Seismic Assessment

Following the equivalent load-displacement method outlined in [19], a seismic assessment was performed for 2D and 3D joints. A comparison of the load–rotation curves, secant and tangent stiffness, equivalent damping and energy dissipation, as defined in [20], was carried out. In Figure 6, a similar cyclic response of the joints studied can be observed. The M2 model reached 66% and 8.8% more load in comparison to the M1 and M3 models, respectively, and the M4 model developed 51% more load than the M3 model. Furthermore, the M1 and M2 models reached a 0.046 (rad) drift ratio, while the M3 and M4 models achieved a 0.065 (rad) drift ratio. Therefore, higher values of rotation were developed by 3D joints than 2D joints, although weak-axis moment connections were used. The response in all joints showed an isotropic behavior, similar to research performed by [21], where hysteretic behavior without pinching or brittle failure mechanisms were exhibited.

Figure 7 shows the evolution of the normalized tangent stiffness (tangent unloading stiffness $K_t$ divided by the initial elastic stiffness $K_o$) with the rotation amplitude of the connection. The M1 and M2 models reached values close to 1.0 for all rotation and axial load levels. However, a decrease in the M3 and M4 models for the 50% axial load case was obtained. Additionally, when the rotation is higher than 2% a reduction of tangent stiffness was observed. Figure 8 shows the normalized secant stiffness evolution. A symmetric response in the M1 model with values around 1.0 until 1% of rotation was obtained and an asymmetric response in the M2 model with a major influence for the 50% axial load case was observed. However, a major decrease in secant stiffness in the M3 and M4 models was obtained when rotations exceeded 1%. Therefore, there may derive an influence of the number of beams connected in the cyclic response.

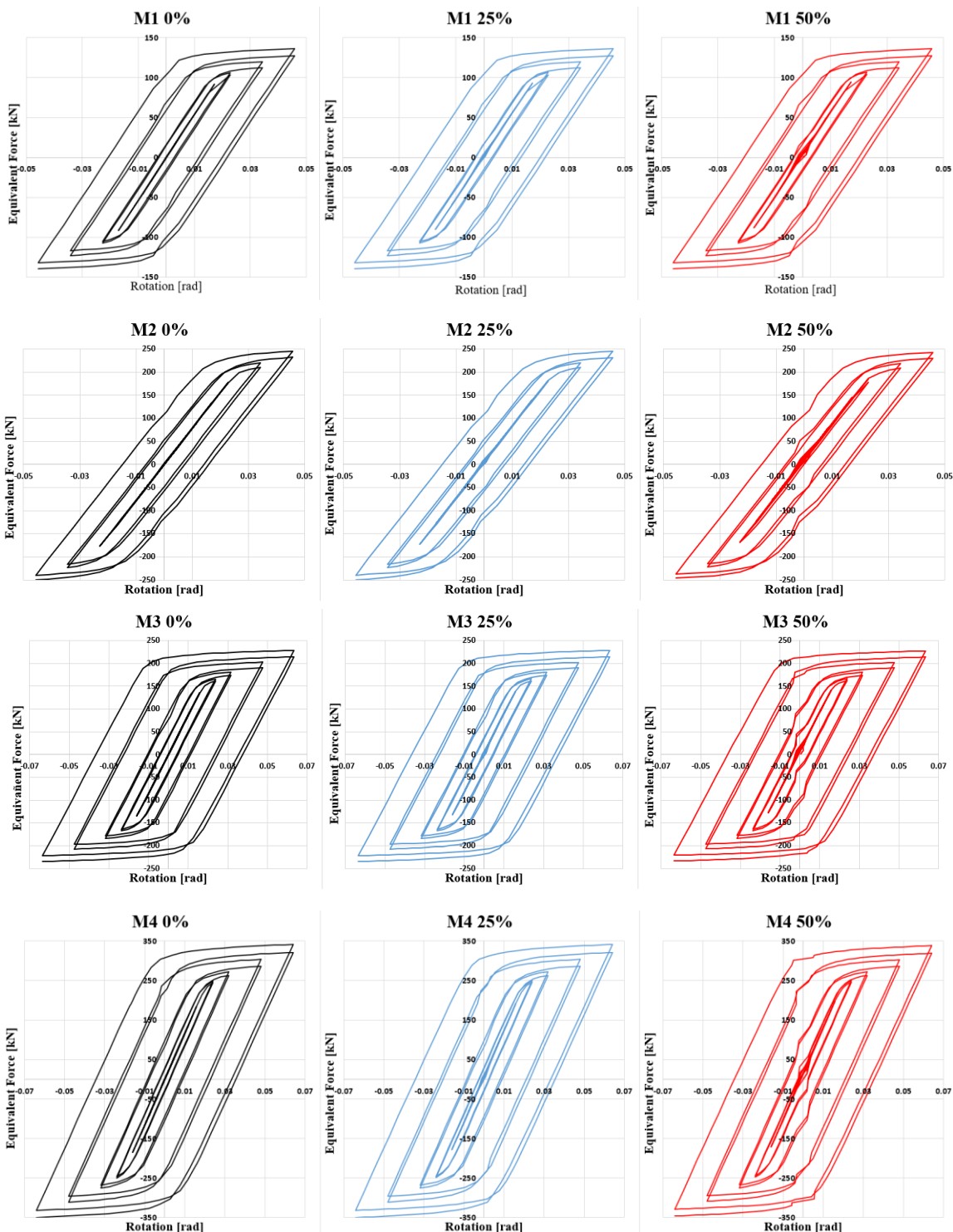

**Figure 6.** Equivalent force-rotation curves for the configurations studied.

In Figure 9, the dissipated energy (DE) is reported. The M3 and M4 models achieved higher values of DE than the M1 and M2 models. Therefore, a relation between the number of beams connected and the DE may be deduced. In Figure 10, the equivalent damping (ED) for 0.04 (rad) drift was higher than 12% in all joints, except in the M2 model which reached 9.5% at the same drift. These values are lower than the values obtained for joints with hollow structural section columns, where a value of 30% was reported according to [13].

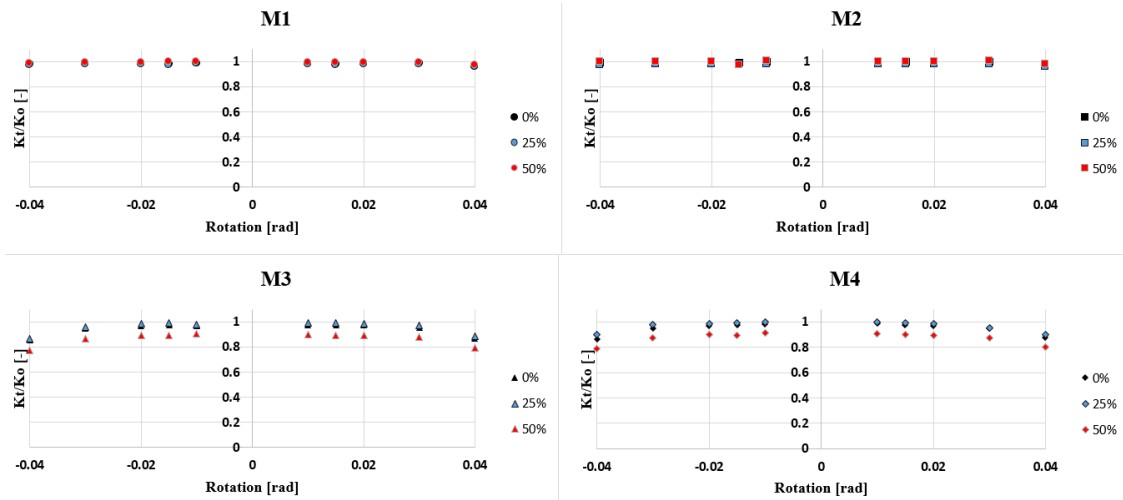

**Figure 7.** Normalized tangent stiffness vs. rotation for the configurations studied.

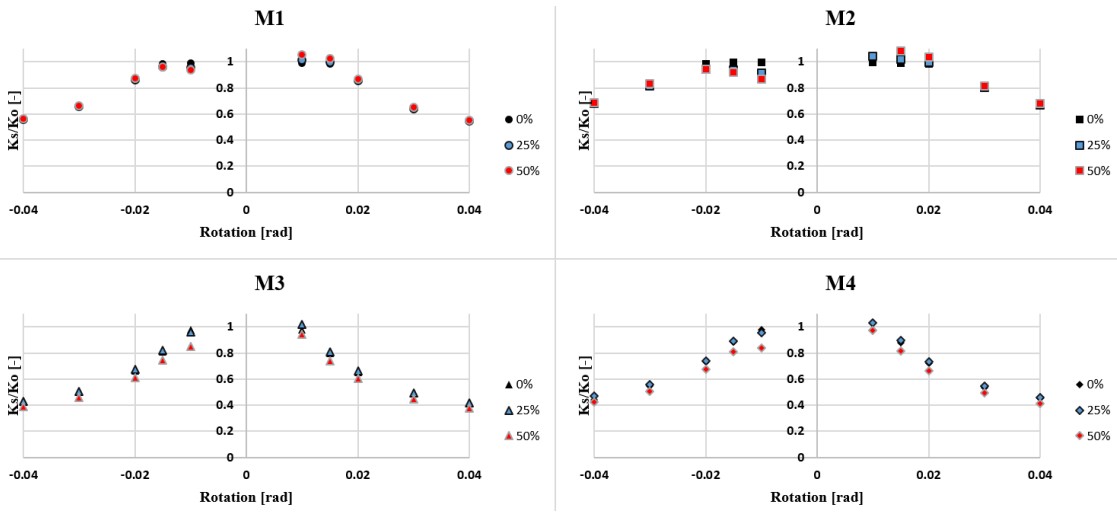

**Figure 8.** Normalized secant stiffness vs. rotation for the configurations studied.

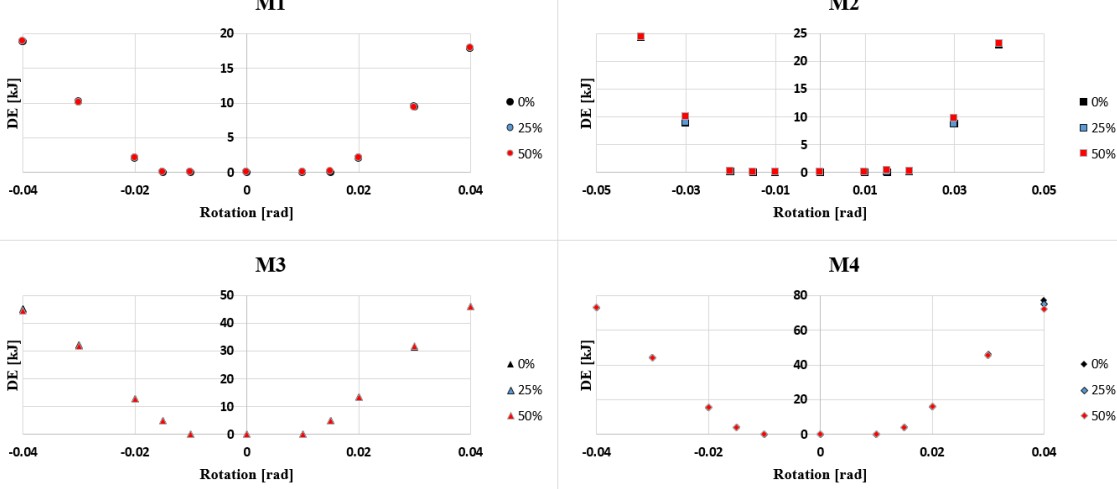

**Figure 9.** Dissipated energy vs. rotation for the configurations studied.

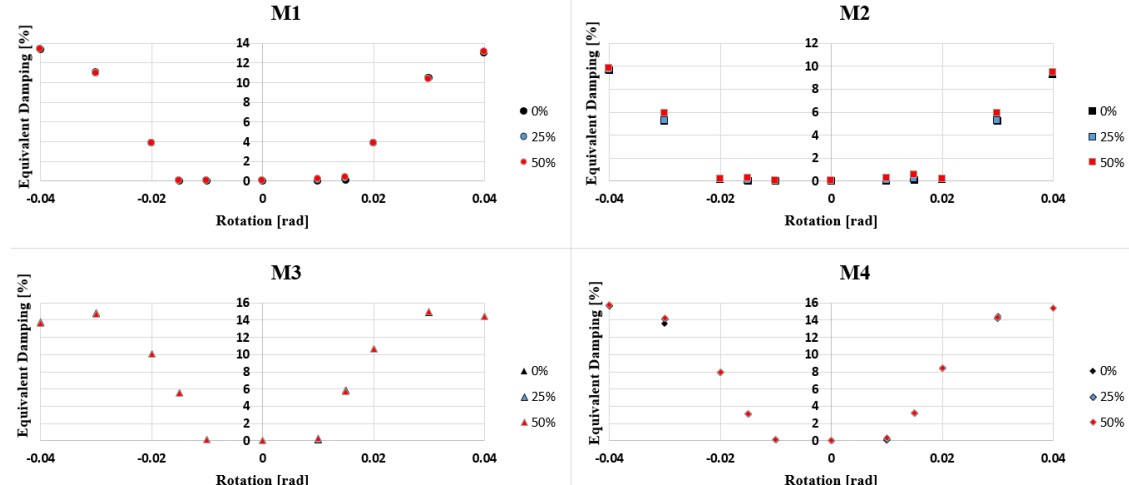

**Figure 10.** Equivalent damping vs. rotation for the configurations studied.

In Figure 11, moment-rotation curves for one beam connected by minor axis and one beam connected by major axis are reported. The moment values were normalized by the expected plastic moment $M_p$ = 486.61 (kN-m) and the rotation values were normalized by a factor equivalent to $M_p/(2\ EI/L)$, where L is the beam length, E is the Young's modulus of steel, and I is the beam moment of inertia. The 20 EI/L and 2 EI/L values are stiffness limits that separate fully restrained, partially restrained and simple connections, according to [22]. Clearly, the major axis connections can be classified as fully restrained. The minor axis connections fall in the partially restrained region. However, the M3 model exhibited a decrease in stiffness as the axial load increased, transitioning from fully restrained to partially restrained. A partially restrained response for all cases was obtained in the M4 model.

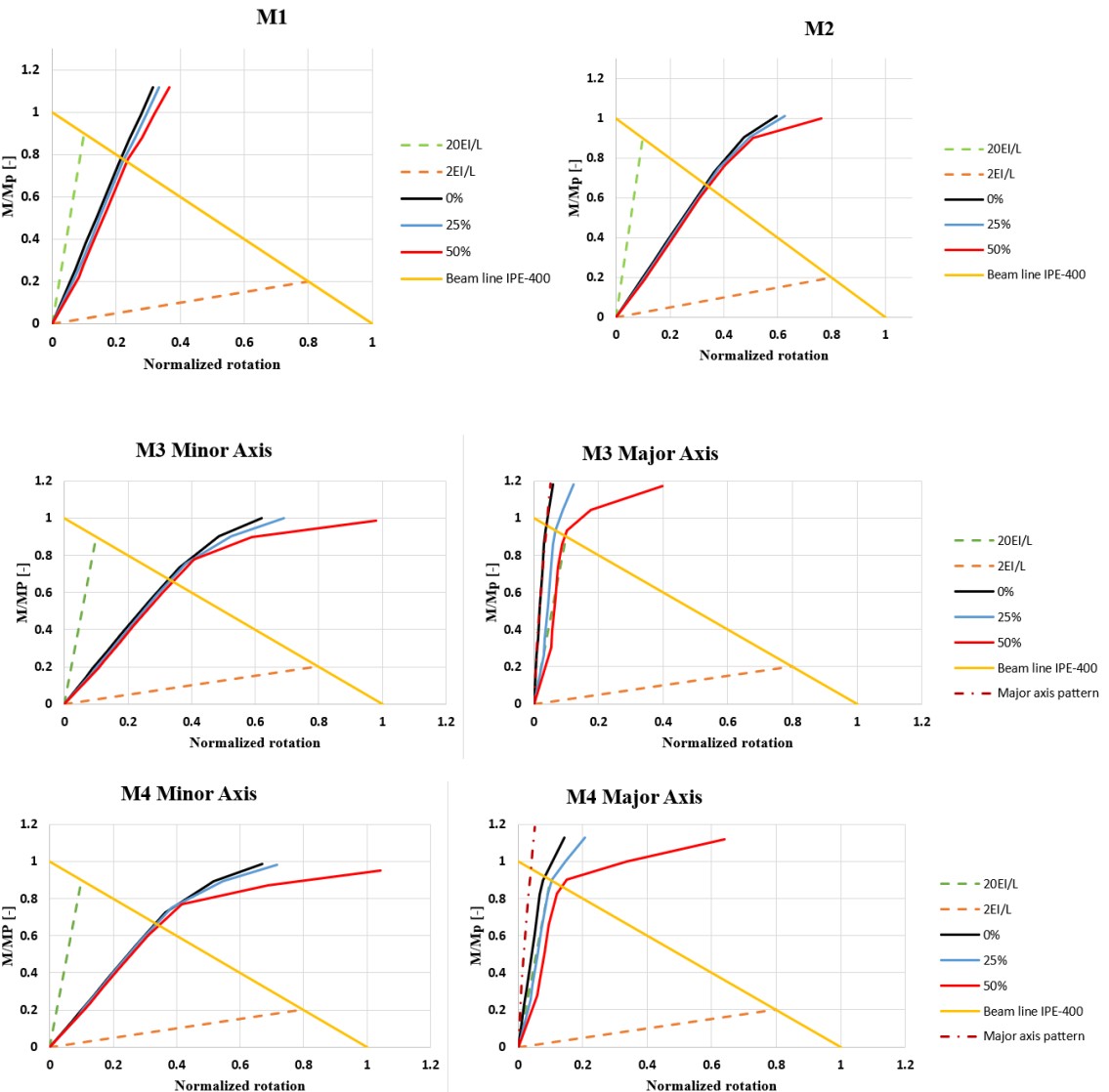

**Figure 11.** Normalized moment-rotation curves, elastic stiffness and comparison with limits established in AISC-360 Specification, data from [22], at each east beam.

## 4.2. Hysteretic Behavior

The M1, M2, M3 and M4 models achieved similar hysteretic behavior. A large number of results were obtained, therefore, only the results of one minor axis connection and one major axis connection are reported. In Figure 12, a 4% drift ratio and flexural strength of the beam greater than 0.80 $M_p$, where $M_p = 486.61324$ (kN-m), were obtained. However, a notable decrease in stiffness in models with minor axis beams was obtained. A behavior without pinching was exhibited, showing the influence of web column rigidity in the cyclic behavior of moment connections.

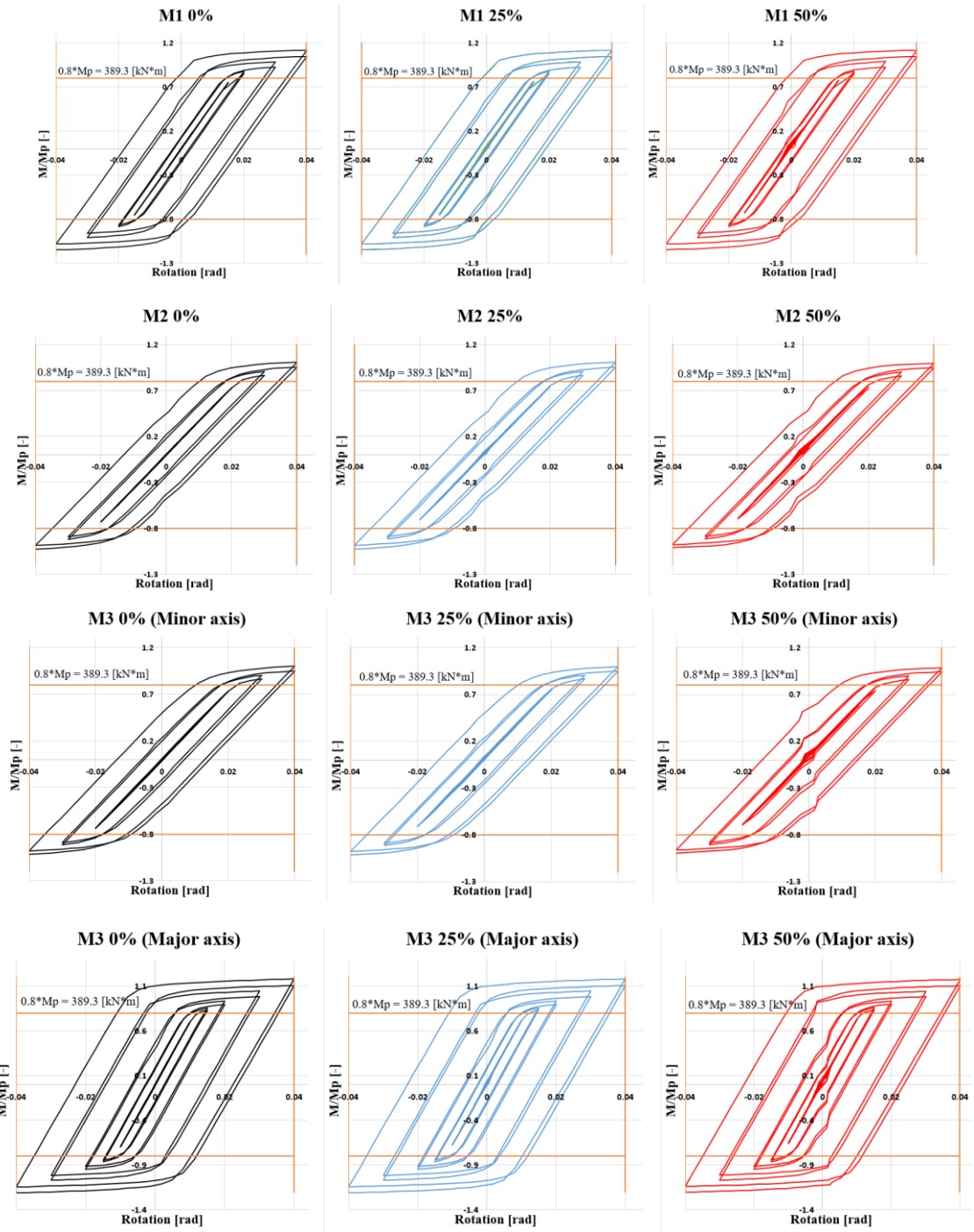

**Figure 12.** *Cont*.

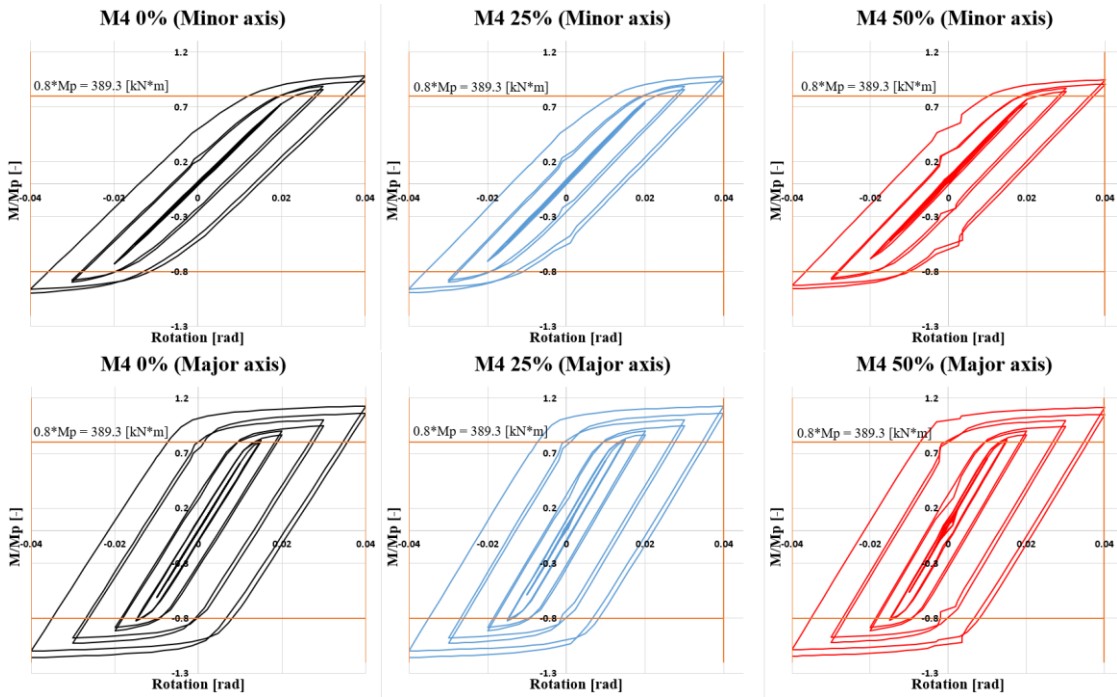

**Figure 12.** Normalized moment-rotation curves at each east beam.

### 4.3. Failure Mechanism

According to [2], a ductile behavior in joints subjected to seismic loads is desirable. Therefore, the formation of plastic hinges in beams before columns is preferred. Plastic hinges in beams and column were reached for the M3 and M4 models. The distribution of von Mises equivalent stress in the final deformation of each model at the maximum load is reported in Figure 13. Plastic strains developed in the beams; however, plastic strains also appeared in the column for the M2, M3 and M4 models at 50% axial load level. A slight inelastic incursion in the column for the 25% axial load case in these models was reached.

Furthermore, the cyclic response of weak-axis moment connections without pinching can be explained by the limited plastic strains reached in the beams up to 4% drift ratio, as a consequence of the low elastic stiffness provided by the weak axis of the column. However, a major elastic stiffness is obtained by strong-axis moment connections and a large plastic strain is mainly developed by the flange and web of the connected beams up to 4% drift ratio, according to results obtained by [23], affecting the hysteretic curve in terms of strength and stiffness. Finally, the effect of initial imperfections can be representative of the cyclic performance of strong-axis moment connections according to results obtained in [24].

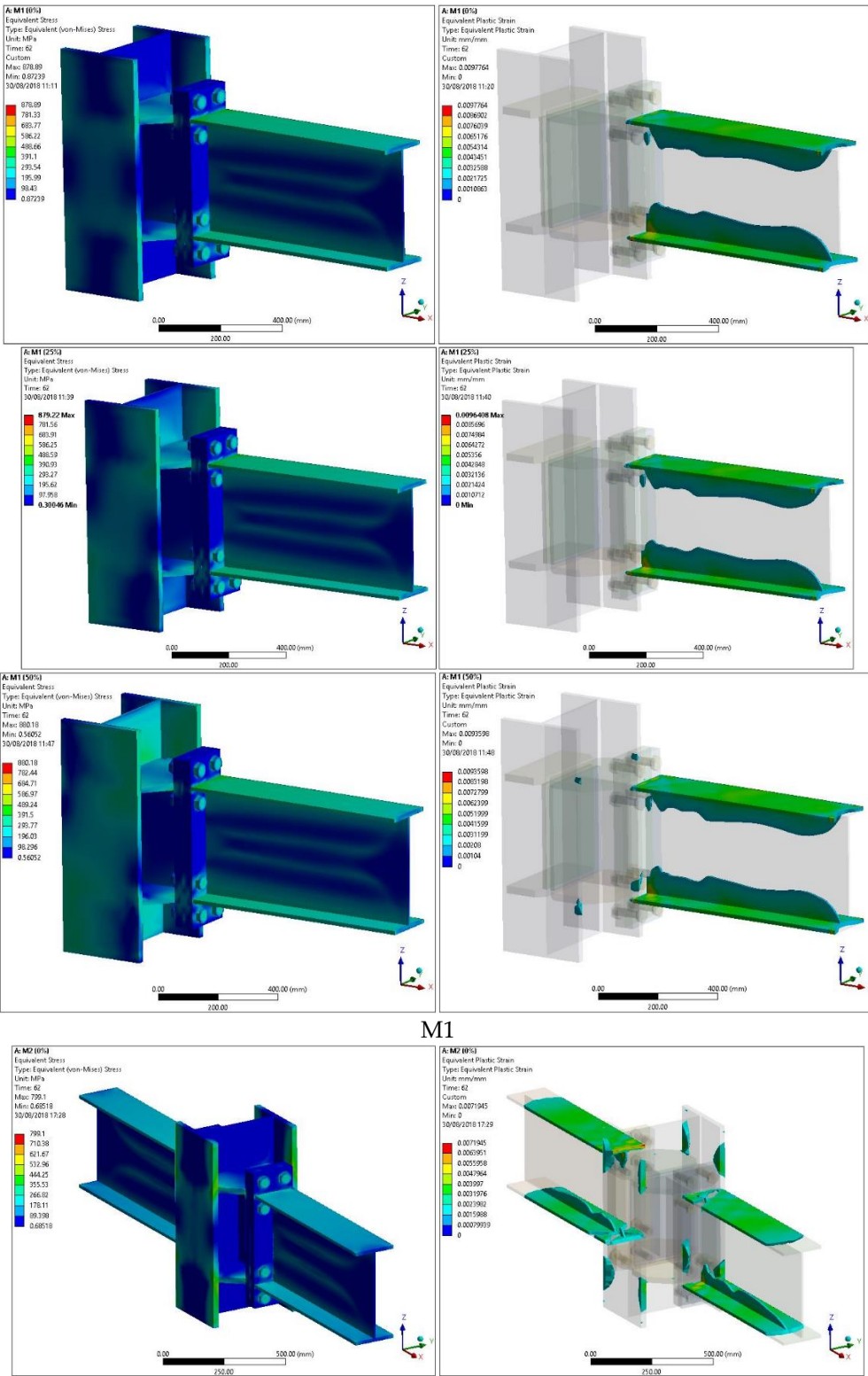

**Figure 13.** *Cont.*

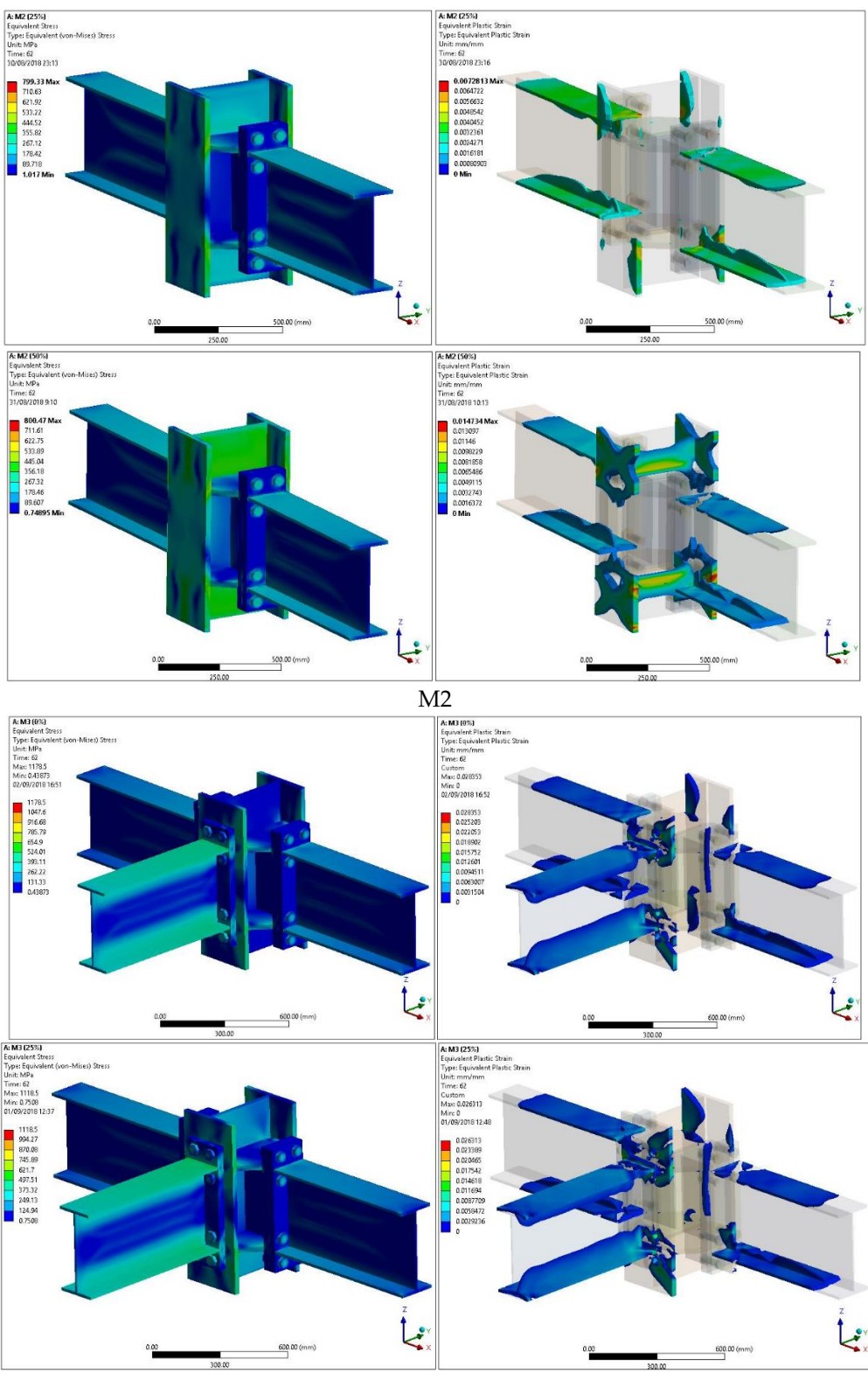

**Figure 13.** *Cont.*

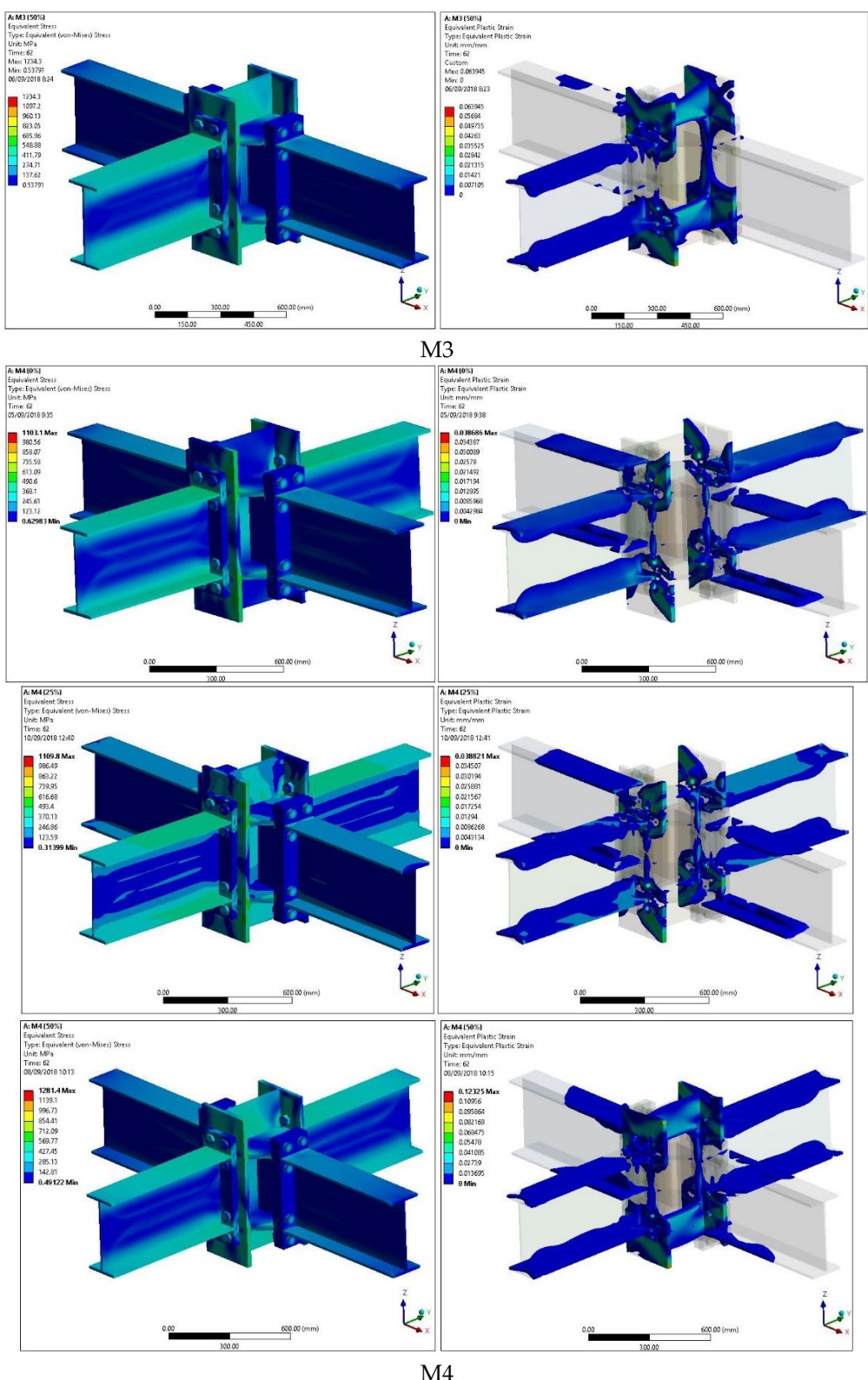

**Figure 13.** Distribution of von Mises stress and plastic strains by applied maximum displacement for the configurations studied (**M1, M2, M3** and **M4**).

## 5. Conclusions

In this research, the bidirectional response of weak-axis end-plate moment connections between I-beams and H-columns was studied. A numerical study based on finite element models of M1 (one beam, exterior), M2 (two beams, interior), M3 (three beams, exterior) and M4 (four beams, interior) connections with different levels of axial load was performed. The results obtained showed that end-plate moment connections with four bolts connected by the weak-axis of columns satisfy the design criteria, failure mechanisms and performance according to Seismic Provisions [2]. However, a partially restrained response was observed, showing the significant influence of weak axis connections to columns in the performance of joints subjected to cyclic loads. In terms of resistance, the elements of the connection remained elastic at 4% of drift ratio. Using the equivalent load-displacement method [19], a comparison of cyclic response was performed, showing that 3D models developed higher deformations than 2D models. Similarly, the equivalent damping and dissipated energy in the 3D models reached higher values than the 2D models. Finally, the elastic stiffness of weak axis moment connections shall be considered in the design of buildings with steel moment frames, which can affect drift verifications. Additionally, it is necessary to verify the strong-column/weak-beam criteria established in [2], considering the beams connected by the weak axis, which may affect the performance of the columns, avoiding an incorrect estimation of the flexural resistance of the columns.

**Author Contributions:** Conceptualization, E.N.; methodology, E.N.; software, G.P., E.N.; validation, E.N. and R.H.; formal analysis, G.P.; investigation, E.N., R.H. and G.P.; writing—original draft preparation, E.N.; writing—review and editing, R.H., E.N.; visualization, R.H.; supervision, R.H. All authors have read and agreed to the published version of the manuscript.

**Funding:** This research received no external funding.

**Conflicts of Interest:** The authors declare no conflict of interest.

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
