# Peer review of "Bidirectional Response of Weak-Axis End-plate Moment Connections: Numerical Approach"

_metals, doi:10.3390/met10070964_

Round 1

Reviewer 1 Report

The topic is interesting for the Readers of this journal. Indeed, the behaviour of 3D joints is a typical problems for many designers. However, if moment frames are designed as planar system the joints are mainly 2D since shear connections are used in trasnvserse direction, Therefore, I suggest to explain better the field of application of this type of joints.

I also recommend to clarify the design criteria of the joints. How did you apply the hiearchy of resistances at local and global level?

Regarding the FEA and the results of the analyses I suggest to explain better the modelling assumptions, e.g. how did you modelled the bolts in tension?Can the models simulate the nut stripping or the shank necking? With this regard, the Authors can consider the following study as potential reference for the discussion (if relevant): D’Aniello M., Cassiano D., Landolfo R., (2017) Simplified criteria for finite element modelling of European preloadable bolts. Steel and Composite Structures, An International Journal Vol. 24, No. 6 (2017) 643-658

The cyclic response of the joint looks very stable, but I have some concerns about the simulated hysteretic shapes. If plastic hinge forms in a member the local buckling of flange produces a loss of resistance. To simulate this effect out-of-square imperfections should be modelled. A way to simulate this effect can be found in Tartaglia R., D’Aniello M., Rassati G.A., Swanson J.A., Landolfo R. (2018). Full strength extended stiffened end-plate joints: AISC vs recent European design criteria. Engineering Structures, Volume 159, 15 March 2018, Pages 155–171. 

Please add some additional discussion on the implications for designon the basis of the obtained results

Author Response

Metals, Special Issue "Advances in Structural Steel Research"

MDPI

Subject: Revised Manuscript “metals-836585”

Dear Editorial Staff:

Please find enclosed the updated version of the paper entitled “Bidirectional Response of Weak-Axis End-plate Moment Connections: Numerical approach" by Eduardo Nuñez, Guillermo Parraguez, and Ricardo Herrera. This draft has been extensively revised to address the comments received following the initial review of the paper. The authors would like to thank the reviewers for their comments, which were very helpful in improving the quality and clarity of the manuscript. Attached is a summary of how the recommendations of the reviewers were addressed in the revised version of the paper.

Should you need to contact me, please contact me via email at [email protected]

Sincerely,

Eduardo Nuñez, PhD

Department of Civil Engineering

Universidad Católica de la Santísima Concepción, Concepción, Chile

Reviewer 2 Report

See attached file "comments to authors"

Author Response

(The authors gave the same response as above.)

Reviewer 3 Report

The authors presented an interesting numerical experiment on the behavior of the flange beam-column joint in steel frames. The behavior of the joint is a fundamental aspect of frames stiffness and resistance. The paper deals with a fundamental aspects, and presents the matter in a impressive and correct way. Some typos and imprecisions are in the text and has been reported in the attached file.

Author Response

(The authors gave the same response as above.)
